# Safety of laparoscopic pancreaticoduodenectomy in patients with liver cirrhosis using propensity score matching

**Ke Cheng, Wei Liu, Jiaying You, Shashi Shah, Yunqiang Cai, Xin Wang, Bing Peng** *

Pancreatic Department, The West China Hospital, Sichuan University, Chengdu, Sichuan, China

* drpengbing123@163.com

**Data Availability Statement:** All relevant data are within the manuscript.

## Abstract

Currently, safety of laparoscopic pancreaticoduodenectomy (LPD) in patients with liver cirrhosis is unknown. The aim of this study was to explore postoperative morbidity and mortality and long-term outcomes of cirrhotic patients after LPD. The study was a one-center retrospective study comprising 353 patients who underwent LPD between October 2010 and December 2019. A total of 28 patients had liver cirrhosis and were paired with 56 non-cirrhotic counterparts through propensity score matching (PSM). Baseline data, intra-operative data, postoperative data, and survival data were collected. Postoperative morbidity was considered as primary outcome whereas postoperative mortality, surgical parameters (operative durations, intraoperative blood loss), and long-term overall survival were secondary outcomes. Cirrhotic patients showed postoperative complication rates of 82% compared with rates of patients in the control group (48%) (P = 0.003). Further, Clavien-Dindo ≥III complication rates of 14% and 11% (P = 0.634), Clavien-Dindo I-II complication rates of 68% and 38% (P = 0.009), hospital mortality of 4% and 2% (P = 0.613) were observed for cirrhotic patients and non-cirrhotic patients, respectively. In addition, an overall survival rate of 32 months and 34.5 months (P = 0.991), intraoperative blood loss of 300 (200–400) ml and 150 (100–250) ml (P<0.0001), drain amount of 2572.5 (1023.8–5275) ml and 1617.5 (907.5–2700) ml (P = 0.048) were observed in the cirrhotic group and control group, respectively. In conclusion, LPD is associated with increased risk of postoperative morbidity in patients with liver cirrhosis. However, the incidence of Clavien-Dindo ≥III complications and post-operative mortality showed no significant increase. In addition, liver cirrhosis showed no correlation with poor overall survival in patients who underwent LPD. These findings imply that liver cirrhosis patients can routinely be considered for LPD at high volume centers with rigorous selection and management.

## Introduction

Pancreaticoduodenectomy (PD) is the conventional surgical approach for managing pancreatic, duodenal neoplasms and other lesions located in the pancreatic head and periampullary

**Funding:** This work was supported by grants from the 1.3.5 project for disciplines of excellence clinical Research Incubation Project, West China Hospital, Sichuan University (No.2018HXFH015).

**Competing interests:** The authors have declared that no competing interests exist.

region [1]. On the other hand, laparoscopic pancreaticoduodenectomy (LPD) is one of the most challenging gastrointestinal operations to perform. Although LPD approach is identical to PD, LPD is characterized by short hospital stays, low blood loss, fewer complications, and similar long-term overall survival outcomes [2,3]. However, LPD is not widely used as a routine surgical approach owing to its technical limitations [4,5].

Liver cirrhosis is a life-threatening health condition that results in multiple severe complications, including variceal bleeding, ascites, spontaneous bacterial peritonitis, and encephalopathy [6]. In most parts of Asia, Hepatitis B virus (HBV) infection is the leading cause of liver cirrhosis [6]. Notably, China has high incidence of liver cirrhosis in Asia which is attributed to the high HBV incidence. Cirrhotic patients present with increased risk of bleeding, infections, hepatic decompensation, and hepatic encephalopathy following a major abdominal surgery [6]. Liver cirrhosis is associated with poor outcomes following intra-abdominal surgery [7]. Previous studies report high postoperative morbidities and mortalities in cirrhotic patients [8,9]. For example, Nguyen *et al* [9] report that higher incidence of postoperative complications in cirrhotic patients (adjusted odds ratio, 1.35; 95% confidence interval, 1.20–1.52) compared with patients with no cirrhosis following colorectal surgery. In addition, colorectal cancer patients with cirrhosis showed significantly higher in-hospital mortality compared with colorectal cancer patients with no cirrhosis (14% vs. 5%, respectively, $P < 0.0001$). Liver cirrhosis was considered a contraindication for pancreatic or biliary tract surgery in the past [10]. Currently, advances in surgical techniques and medical management have resulted in development of safe and effective surgical procedures for use in cirrhotic patients [11,12]. Previous studies report that laparoscopic surgery is a safe and less invasive alternative to open surgery for cirrhotic patients [13,14]. For example, Kim *et al* [15] reported high efficacy and safety of laparoscopic gastrectomy (LG) in gastric cancer patients with liver cirrhosis. In addition, Zhou *et al* [13] report that laparoscopic colorectal surgery is a safe and less invasive alternative for open surgery in some cirrhotic patients.

Although a few previous studies reported on outcomes of PD in patients with liver cirrhosis, the safety of LPD procedure has not been explored previously [16–19]. PD and LPD procedures have several differences; therefore, it is necessary to explore short- and long-term outcomes of LPD in patients with liver cirrhosis. The aim of this retrospective cohort study was, therefore, to evaluate the effect of liver cirrhosis on postoperative morbidity, mortality, and overall survival outcomes in patients who underwent LPD in our center. The findings of this study will provide information of the safety of LPD procedure for cirrhotic patients.

## Materials and methods

### Study design

We retrospectively evaluated patients who underwent LPD procedure at the Pancreatic Department, West China Hospital, between October 2010 and December 2019. All data were collected in May 2020 and were fully anonymized before we accessed them. Identity of individual participants was blinded during or after data collection. Patients diagnosed with resectable malignant tumors or benign diseases located in the periampullary or pancreas, who underwent LPD procedure were included in the study. Exclusion criteria comprised patients who were postoperatively treated in other medical centers or underwent laparotomy procedure, patients with poor liver function, and patients with thrombosed portal veins, malnutrition, and severe malignant tumors (patients with locally advanced, metastatic, or inoperable advanced cancer). Diagnosis of cirrhosis was carried out through ultrasound, CT examination, and intra-operative findings or liver biopsies.

Medical records for all the patients were reviewed then each cirrhotic patient was paired with two non-cirrhotic counterparts using PSM method [20] to minimize effect of confounders such as age, gender, body mass index (BMI), and diagnosis of diseases. Post-operative morbidity was considered as the primary outcome whereas mortality, surgical parameters, and long-term overall survival were secondary outcomes. Patients signed the informed consent that their clinic data will be used in medical study before admitted to hospital. The study was approved by the ethics committee of Sichuan University.

## Preoperative assessment

Demographic data for each patient was collected, including age, gender, BMI, diagnosis, American society of anesthesiologists classification (ASA) [21], and laboratory tests (blood tests, liver and kidney function tests, and coagulation function tests). In addition, liver function for the cirrhotic patients was graded using the Child-Pugh classification [22] and Model for End-stage Liver Disease (MELD) score [23]. Further, a revaluation of Child-Pugh class was performed after biliary drainage.

## Operative data

All LPD operations were performed by the same group of surgeons using standard procedures described previously by [24]. Operative information including operative time, blood loss, blood transfusion, pancreatic texture and duct diameter, site of the tumor, tumor size, and vascular resection were collected.

## Postoperative data

Drain was removed from all patients in cases where no bile, pancreatic, or other leaks were present. Postoperative laboratory tests were carried out after 3 postoperative days (POD). Number of hospital mortalities, 90-day mortalities, incidence of reoperation, period of ICU, hospital stay, drain removed and amount of drain, and postoperative complications, were recorded.

## Definitions

All postoperative complications were stratified using Clavien-Dindo classification of surgical complications [25]. Postoperative pancreatic fistula (POPF), delayed gastric emptying (DGE), and post-pancreatectomy hemorrhage bleeding (PPH) were defined according to the International Study Group of Pancreatic Fistula (ISGPF) guidelines [26–28]. Postoperative ascites was defined as effusion of more than 400ml/d through the drain after POD 4 [29]. Reoperation was defined as a secondary operation due to severe complications following LPD. Hospital mortality was defined as death within 30 days following surgery or during primary hospitalization.

## Follow-up

Follow-up examinations (blood tests, liver and kidney function tests, abdominal CT scans) were conducted at 3-months intervals for the first 2 years after LPD. Follow-up examinations were changed to a 6-months interval for patients who exhibited no signs of recurrence 2 years after surgery. Follow-up data were collected by examining outpatient medical records and through personal communication by telephone.

## Statistical analysis

PSM was calculated using logistic regression to minimize potential bias between the two groups. A 2:1 patient matching was performed using nearest-neighbor matching method without replacement. A caliper radius equal to a standard deviation of 0.1 was set to prevent poor matching. Variables included in the matching model were gender, age, BMI, and diagnosis.

For continuous variables, results were reported as means ± standard deviations of the mean (SD) for normally-distributed data. Median (25 quantile-75 quantiles) was reported for non-normally distributed data. Frequencies and percentages were reported for categorical variables. Categorical variables were analyzed using Chi-square test. Continuous variables were analyzed using Student *t*-test for normally distributed data, whereas non-parametric test was used for non-normally distributed data. Kaplan-Meier method [30] was used to calculate overall survival, using GraphPad Prism (version 8, GraphPad Software, San Diego, USA), and results compared by the log-rank test. Values with P < 0.05 were considered statistically significant by the two-tailed test. All statistical analyses were performed using Statistical Product and Service Solutions (SPSS) statistical software (version 26, IBM Corporation, New York, USA).

## Results

### Patient characteristics

A total of 353 patients met the study inclusion criteria. Diagnosis of pancreatic carcinoma was significantly different between the two cohort groups before PSM (P = 0.03). After PSM, the matched cohort comprised 28 patients in the cirrhotic group and 56 patients in the control group, and bias factors were modified (Table 1). All results of the study were based on PSM analysis. In the cirrhotic group, 15 (54%) patients were diagnosed with pancreatic carcinoma, 3 (11%) had chronic pancreatitis, 9 (32%) exhibited periampullary carcinoma whereas 1 (4%) patient had another disease. In the control group, 30 (54%) patients were diagnosed with pancreatic carcinoma, 20 (36%) hand periampullary carcinoma, 2 (4%) exhibited chronic pancreatitis whereas 4 (7%) patients had other benign or borderline pancreas tumors. In addition, 11 (39%) patients were diagnosed with Child-Pugh A liver function, 17 (61%) patients were diagnosed with Child-Pugh B liver function and none was diagnosed with Child-Pugh C liver function. Median MELD score was 12 (8–17). The cirrhotic group showed no statistical difference in baseline conditions compared with the control group. In the cirrhotic group, 16 (57%)

**Table 1. Comparison of baseline characteristics between the two groups before and after PSM.**

| | Before PSM | | | After PSM | | |
|---|---|---|---|---|---|---|
| Characteristic | Cirrhosis(n = 28) | Control(n = 325) | P-value | Cirrhosis(n = 28) | Control(n = 56) | P-value |
| **Gender** | | | 0.098 | | | 1 |
| Male | 21[75] | 192[59] | | 21[75] | 42[75] | |
| Female | 7[25] | 133[41] | | 7[25] | 14[25] | |
| **Age (years)** | 62 (50–70.5) | 62 (51–70) | 0.995 | 62 (50–70.5) | 60 (52–71) | 0.798 |
| **BMI (kg/m$^2$)** | 21.4±2.1 | 21.7±2.9 | 0.635 | 21.4±2.1 | 21.5±1.7 | 0.686 |
| **Diagnosis** | | | | | | |
| Pancreatic carcinoma | 15[54] | 108[33] | **0.03** | 15[54] | 30[54] | 1 |
| Periampullary carcinoma | 9[32] | 133[41] | 0.363 | 9[32] | 20[36] | 0.746 |
| Other diseases | 4[14] | 84[26] | 0.175 | 4[14] | 6[11] | 0.905 |

BMI: Body mass index, PSM: Propensity score matching.

**Table 2. Baseline data of patients in cirrhotic and control groups.**

| | Patients with liver cirrhosis [%] | Control patients [%] | P-value |
|---|---|---|---|
| **Patients** | 28 | 56 | |
| **Gender** | | | 1 |
| Male | 21[75] | 42[75] | |
| Female | 7[25] | 14[25] | |
| **Age (years)** | 62 (50–70.5) | 60 (52–71) | 0.798 |
| **BMI (kg/m$^2$)** | 21.4±2.1 | 21.5±1.7 | 0.686 |
| **Diagnosis** | | | |
| Pancreatic carcinoma | 15[54] | 30[54] | 1 |
| Chronic pancreatitis | 3[11] | 2[4] | 0.415 |
| Periampullary carcinoma | 9[32] | 20[36] | 0.746 |
| Other diseases | 1[4] | 4[7] | 0.87 |
| **ASA score** | | | 0.19 |
| II | 16[57] | 40[71] | |
| III | 12[43] | 16[29] | |
| **Cause of liver cirrhosis** | | | |
| HBV | 13[46] | | |
| Alcohol abuse | 4[14] | | |
| Biliary-oriental cirrhosis | 7[25] | | |
| Schistosomiasis | 1[4] | | |
| Cryptogenic cirrhosis | 3[11] | | |
| **Child-Pugh classification** | | | |
| A | 11[39] | | |
| B | 17[61] | | |
| **MELD score** | 12(8–17) | | |
| **Preoperative laboratory Examinations** | | | |
| Blood platelet (10$^9$) | 188 (133.3–278) | 206 (147.3–261.8) | 0.491 |
| RBC (10$^{12}$) | 3.9±0.6 | 4.1±0.5 | 0.125 |
| WBC (10$^{12}$) | 5.9 (3.9–8.1) | 5 (4–6.6) | 0.273 |
| Albumin(g/L) | 37.3±5.4 | 38.9±5.7 | 0.205 |
| Total bilirubin(μmol/L) | 93.6 (14.9–186.5) | 77.7 (19.8–208.9) | 0.824 |
| Direct bilirubin (μmol/L) | 84.6 (5.4–172.6) | 69.3 (9.4–191.1) | 0.827 |
| ALT (g/L) | 88 (24.8–212.3) | 78.5 (25.4–150) | 0.462 |
| AST (g/L) | 86 (26–111.8) | 66 (23–156.8) | 0.876 |
| Serum creatinine (μmol/L) | 67.5 (62–78.5) | 72 (62.5–81.8) | 0.217 |
| PT(s) | 11.9 (11–12.7) | 11.8 (11.2–12.9) | 0.831 |
| INR | 1.04 (0.95–1.11) | 1.03 (0.97–1.11) | 0.754 |

BMI: Body mass index, ASA: American society of anesthesiologists, HBV: Hepatitis B virus, MELD: Model for End-stage Liver Disease, ALT: Alanine transaminase, AST: Aspartate aminotransferase, PT: Prothrombin time, INR: International normalized ratio, RBC: Red blood cell, WBC: White blood cell.

patients were ASA II, whereas 12 (43%) were ASA III, however, no significant difference was observed between the two groups (P = 0.19). Preoperative laboratory examinations showed no significant differences between the two groups. All preoperative results are summarized in Table 2.

**Table 3. Operative data and characteristics of patients in cirrhotic and control groups.**

| | Patients with liver cirrhosis [%] (n = 28) | Control patients [%] (n = 56) | P-value |
|---|---|---|---|
| Operative time (min) | 380.7±84.4 | 352.3±93.3 | 0.173 |
| Blood loss (ml) | 300 (200–400) | 150 (100–250) | **<0.0001** |
| Blood transfusion | 2[7] | 3[5] | 0.744 |
| Pancreatic texture | | | 0.636 |
| Soft | 10[36] | 23[41] | |
| Firm | 18[64] | 33[59] | |
| Pancreatic duct diameter (mm) | 4(3–5.5) | 3(3–4) | 0.127 |
| Site of tumor | 25 | 54 | |
| Ampulla | 0[0] | 4[7] | 0.398 |
| Duodenum | 7[28] | 10[19] | 0.34 |
| Common bile | 2[8] | 6[11] | 0.98 |
| Pancreas | 16[64] | 34[63] | 0.929 |
| Tumor size (cm) | 2.5 (1.6–3.4) | 3 (2.2–4.5) | 0.175 |
| Vascular resection | 5[18] | 13[23] | 0.573 |

## Operative data

Tumor size, pancreatic texture, duct diameter, and operative time showed no significant differences between the two groups. However, the control group showed significantly lower median intraoperative blood loss of 150 (100–250) ml compared with median intraoperative blood loss of 300 (200–400) ml observed for the cirrhotic group (P<0.0001). Number of patients receiving an intraoperative transfusion was not significantly different between the two groups. Operative evaluation results are summarized in Table 3.

## Postoperative data

Overall hospital (P = 0.613) and 90-day (P = 0.163) mortalities were not significantly different between the two groups. Similarly, reoperation rate was not significantly different between the two groups (P = 0.793). Cirrhotic patients showed significantly higher drain amounts compared with the amount for the control group, with 2572.5 (1023.8–5275) ml and 1617.5 (907.5–2700) ml, respectively (P = 0.048). However, drain removal days were not significantly different between the two groups (P = 0.293). Furthermore, duration of ICU and hospital stays were not significantly different between the groups. Incidence of overall postoperative complications were 82% and 48% (P = 0.003) in the cirrhotic and control groups, respectively. Morbidities of complications in Clavien-Dindo I-II were 68% and 38% (P = 0.009) in the cirrhotic and control groups, respectively. Severe complication rates in Clavien-Dindo ≥III were 14% and 11% (P = 0.634) for cirrhotic group and control group, respectively. In addition, patients in the cirrhotic group showed higher pulmonary infection rate compared with pulmonary infection rate of patients in the control group (P = 0.003). Details on complications are listed in Table 4. Cirrhosis group showed higher incidence of Clavien-Dindo I-II complications (P = 0.016) in Child B class compared with patients in the control group. Details on Child-Pugh classification are summarized in Table 5.

In the cirrhotic group, a total of 12 (43%) patients developed postoperative ascites, 4 (14%) showed deterioration of hepatic function and 1 (4%) exhibited upper gastrointestinal tract bleeding. According to Child-Pugh classification, postoperative ascites, deterioration of

**Table 4. Postoperative outcome of patients in cirrhotic and control groups.**

| | Patients with liver cirrhosis [%] (n = 28) | Control patients [%] (n = 56) | P-value |
|---|---|---|---|
| **Hospital mortality** | 1[4] | 1[2] | 0.613 |
| **90-day mortality** | 4[14] | 3[5] | 0.163 |
| **Reoperation** | 3[11] | 5[9] | 0.793 |
| **ICU stay** | 25[89] | 51[91] | 0.793 |
| **Days on ICU** | 1 (1–2) | 1 (1–2) | 0.58 |
| **Hospital days** | 26 (23–35) | 23.5 (20–31.8) | 0.114 |
| **Drain remove days** | 10.5 (7–14.8) | 8 (7–12) | 0.293 |
| **Drain amount (ml)** | 2572.5 (1023.8–5275) | 1617.5 (907.5–2700) | **0.048** |
| **Postoperative complications** | 23[82] | 27[48] | **0.003** |
| Clavien-Dindo I-II | 19[68] | 21[38] | **0.009** |
| Clavien-Dindo ≥III | 4[14] | 6[11] | 0.634 |
| Pancreatic fistula | 8[29] | 10[18] | 0.259 |
| A | 4[14] | 6[11] | 0.905 |
| B | 4[14] | 2[4] | 0.178 |
| C | 0 | 2[4] | 0.55 |
| Delayed gastric emptying | 8[27] | 9[16] | 0.179 |
| Biliary leakage | 1[4] | 1[2] | 0.613 |
| Anastomotic fistula | 0 | 0 | |
| Lymphatic fistula | 6[21] | 5[9] | 0.109 |
| Post-pancreatectomy hemorrhage bleeding | 2[7] | 1[2] | 0.212 |
| Abdominal infection | 5[18] | 5[9] | 0.234 |
| Wound infection | 2[7] | 2[4] | 0.469 |
| Pulmonary infection | 10[36] | 5[9] | **0.003** |
| **Postoperative laboratory examinations** | | | |
| Blood platelet ($10^9$) | 176 (120.8–225.8) | 163.5 (127.5–239.8) | 0.761 |
| Albumin (g/L) | 28.7±3.9 | 28.1±3.8 | 0.498 |
| Total bilirubin (μmol/L) | 46.2 (22.2–122) | 42.5 (22–98.2) | 0.928 |
| Direct bilirubin (μmol/L) | 42.1 (10.2–107.3) | 32 (12.1–89.1) | 0.887 |
| ALT (g/L) | 44.5 (28.3–88.3) | 34 (18.8–95) | 0.159 |
| AST (g/L) | 31.5 (20–60.3) | 28.5 (17.5–81.8) | 0.417 |
| Serum creatinine (μmol/L) | 64 (48.3–72.5) | 60 (49–74) | 0.237 |
| PT (s) | 12.7 (11.4–13.3) | 12.1 (11.5–12.9) | 0.491 |
| INR | 1.09 (1.01–1.13) | 1.07 (1–1.12) | 0.54 |

ICU: Intensive care unit, ALT: Alanine transaminase, AST: Aspartate aminotransferase, PT: Prothrombin time, INR: International normalized ratio.

hepatic function, and upper gastrointestinal tract bleeding were not significantly different between Child A and Child B classes. MELD scores for Child A and Child B groups were 7 (6–9) and 16 (13.5–18) (P<0.0001), respectively (Table 6).

## Survival data

Kaplan-Meier method was used to analyze data for patients with malignant diseases. A total of 25 (34%) patients died during follow-up. Of the 25 patients, 7 patients were in the cirrhotic group and 18 patients in the control group. However, overall survival rate of cirrhotic patients (32 months) was no significantly different compared with OS of patients in the control group (34.5 months) (P = 0.991, Fig 1).

**Table 5. Outcomes of cirrhotic patients with Child classification.**

|  | Child A | Control | P-value | Child B | Control | P-value |
|---|---|---|---|---|---|---|
| Patients | 11 | 56 |  | 17 | 56 |  |
| **ASA score** |  |  | 0.454 |  |  | 0.327 |
| II | 6[54] | 40[71] |  | 10[59] | 40[71] |  |
| III | 5[42] | 16[29] |  | 7[41] | 16[29] |  |
| MELD score | 7(6–9) |  |  | 16(13.5–18) |  |  |
| **Complications related to pancreatic surgery** |  |  |  |  |  |  |
| Clavien-Dindo I-II | 7[64] | 21[38] | 0.203 | 12[71] | 21[38] | **0.016** |
| Clavien-Dindo≥III | 1[9] | 6[11] | 0.87 | 3[18] | 6[11] | 0.734 |
| Pancreatic fistula | 1[9] | 10[18] | 0.785 | 7[41] | 10[18] | 0.096 |
| Delayed gastric emptying | 3[27] | 9[16] | 0.649 | 5[29] | 9[16] | 0.383 |
| Biliary leakage | 1[9] | 1[2] | 0.263 | 0 | 1[2] | 0.465 |
| Lymphatic fistula | 3[27] | 5[9] | 0.228 | 3[18] | 5[9] | 0.572 |
| Intra-abdominal bleeding | 2[18] | 1[2] | 0.068 | 0 | 1[2] | 0.767 |
| Abdominal infection | 1[9] | 5[9] | 0.675 | 4[24] | 5[9] | 0.237 |
| Wound infection | 1[9] | 2[4] | 0.421 | 1[6] | 2[4] | 0.554 |

ASA: American society of anesthesiologists, MELD: Model for End-stage Liver Disease.

## Discussion

This study explores the safety of LPD in patients with liver cirrhosis. The findings show that Clavien-Dindo ≥III complication rates and hospital mortality of patients in the cirrhosis group were not significantly different from the rates for patients in the control group. Although liver cirrhosis results in higher incidence of overall postoperative complications, patients with pancreatic and periampullary diseases, and liver cirrhosis can undergo LPD at high volume centers with rigorous selection and management.

LPD is a complicated procedure associated with less blood loss, less complication rate, and similar overall survival rates compared with survival rates observed after PD [3]. However, the procedure can only be routinely applied at high volume centers, and by experienced surgeons, as it requires minimal invasive pancreatic operation skills and a steep learning curve [31]. Currently, improvements have been made in LPD operative technique, therefore, operative time is not a drawback for LPD. The approach has seen increased application, with Delitto *et al* [32] reporting 52 cases of LPD in 2017 at a mean operative time of 361 minutes. In our study, the mean operative time for the control group was 352.3 minutes. These findings imply that

**Table 6. Comparison of liver cirrhosis relative outcomes of patients in Child A and Child B classes.**

|  | Child A | Child B | P-value |
|---|---|---|---|
| Patients | 11 | 17 |  |
| MELD score | 7(6–9) | 16(13.5–18) | **<0.0001** |
| **Complications related to cirrhosis** |  |  |  |
| Ascites | 3[27] | 9[53] | 0.18 |
| Deterioration of hepatic function | 2[18] | 2[12] | 0.636 |
| Upper gastrointestinal tract bleeding | 1[9] | 0 | 0.206 |

MELD: Model for End-stage Liver Disease.

## Overall survival (n=74)

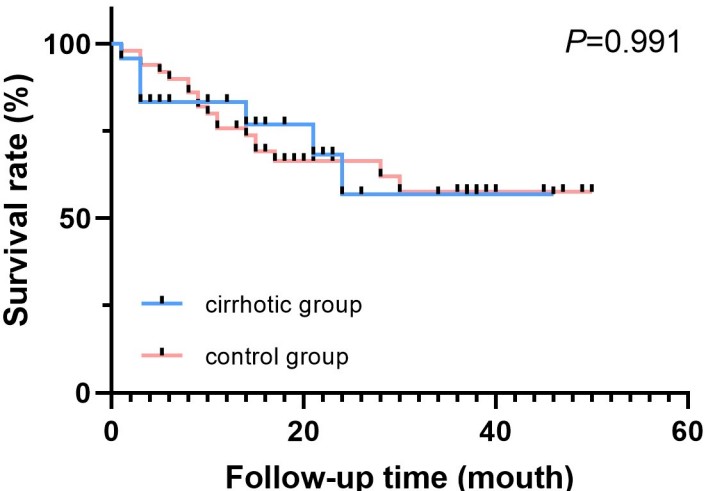

**Fig 1. Overall survival of cirrhotic patients and control patients (patients with malignant diseases).**

cirrhosis is not a risk factor for longer operative time. Several studies report the effect of cirrhosis on the surgical outcome of PD [16–19]. However, currently, no reports on post-operation morbidity and mortality of patients with liver cirrhosis after undergoing LPD.

Patients with liver cirrhosis show a poor response to surgical stress due to loss of liver reserves and other systemic derangements that result from hepatic dysfunction. Incidence of liver cirrhosis patients included in this study, who underwent LPD was 7.93%. Other studies report that between 1.9 and 15.2% of patients receiving PD are diagnosed with liver cirrhosis [16–19].

Laparoscopic techniques can improve outcomes of cirrhosis patients [33]. For example, EI Nakeeb [19] *et al* reported that wound complications, pancreatic fistula, and hospital mortality were significantly higher in cirrhotic patients compared with the control group. Similarly, Regimbeau *et al* [18] reported a higher postoperative morbidity rate of patients in the cirrhotic group compared with patients in the control group (86% and 43%, P<0.001). In addition, postoperative mortality rates in cirrhotic patients and patients in the control group were 17 and 5% (P = 0.94), respectively. In the current study, hospital mortality rates were 4 and 2% (P = 0.613) for patients in cirrhotic and control groups, respectively, which was lower compared with rates reported previously on PD. A previous study comprising 550 patients at our center, reported hospital mortality rate less than 1% after LPD procedure [34]. These low rates can be attributed to the minimally invasive nature of LPD. In our study, only one patient in the cirrhotic group succumbed to postoperative hepatic function failure and pulmonary infection. Complications related to pancreatic surgery were graded using the Clavien-Dindo classification. Severe complications (Clavien-Dindo≥III) in the cirrhotic group were not significantly different compared with complications in the control groups. These findings were different from findings from related studies on PD [17,18]. These findings imply that LPD, a minimally-invasive procedure is safer for cirrhotic patients compared to PD procedure.

Patients with chronic liver diseases are at high risk of contracting infections, due to increased bacterial translocation and immune suppression associated with decreased liver function [35]. For instance, pneumonia is a common infectious disease in cirrhotic patients [36,37]. In the current study, cirrhotic patients showed significantly higher morbidity of postoperative pulmonary infection compared with patients in the control group. This finding implies that cirrhotic patients should be encouraged to go for check-ups when they present with coughing, sputum expulsion, and initiation of deep breathing and take antibiotics, to reduce the risk of pulmonary infection [38]. Furthermore, cirrhotic patients showed significantly higher intraoperative blood loss compared with patients in the control group. High intraoperative blood loss in cirrhotic patients can be attributed to increased bleeding rate and portal hypertension [33]. EI Nakeeb et al [19] reported a median blood loss of 500 ml in the cirrhotic group compared with 200 ml mean blood loss in the control group (P = 0.0001). In the current study, amount of postoperative drain in cirrhotic patients was significantly higher compared with amount of drain in the control group. Higher drain amount in cirrhotic patients can be attributed to postoperative ascites and lower levels of albumin.

Child-Pugh classification and MELD scores were used to assess preoperative hepatic function. However, in our study, not every patient with obstructive jaundice was treated with standardized biliary drainage during the long study period, which may have affected assessment of Child-Pugh classification and MELD score. Moreover, median MELD score of 12 (8–17) was recorded, which may be increased artificially in patients with pancreatic disease due to increased bile duct obstruction and does not directly reflect liver function disorders. Therefore, critical MELD scores may not be effective for prediction of poor outcomes in patients with LPD. Long-term outcomes of cirrhotic patients who underwent LPD showed no significant differences in mean survival rate compared with mean survival rates of patients in the control group. This finding indicates that liver cirrhosis does not result in poor overall survival outcomes in patients with LPD. Notably, similar findings were reported in previous studies [18,19], although inclusion of patients with different diseases in our study, may have affected estimation of overall survival. Further, cirrhotic patients did not have prolonged hospital and ICU stay times compared with patients in the control group. This finding was contrary to findings from previous studies on cirrhotic patients who underwent PD. Warnick et al [17] report longer ICU stay times for cirrhotic patients compared with patients in the control group after PD procedure. In addition, EI Nakeeb et al [19] report longer hospital stay and significantly longer ICU stay after PD procedure for cirrhotic patients compared with patients in the control group. These findings imply that LPD is a less invasive procedure, with less ICU stay time compared with PD procedure for patients with liver cirrhosis.

Our study had some limitations. Firstly, a retrospective design was used and the sample size was small. Secondly, although patients with liver cirrhosis in Child-Pugh B class safely underwent LPD in our study, caution is advised before drawing conclusions because this is one single-center study and preoperative arrangement of obstructive jaundice were not fully standardized. Thirdly, portal hypertension is a major complication of liver cirrhosis. However, portal hypertension was not analyzed in this study due to lack of standardized assessment procedure for portal hypertension.

Fourthly, considering long periods and different types of tumors included, the TNM stage is hard to normalize. Further study focused on one single type of tumor based on the latest version of AJCC guideline could offer more powerful evidence on the overall survival. Besides, multicenter prospective studies with standardized inclusion criteria should be carried out to validate our findings.

## Conclusion

In conclusion, LPD is associated with increased risk of postoperative morbidity in patients with liver cirrhosis. However, incidence of Clavien-Dindo ≥III complications and postoperative mortality were not significantly different from incidence in the control group. In addition, liver cirrhosis was not associated with poor overall survival in patients who underwent LPD. Patients with pancreatic and periampullary diseases, and liver cirrhosis patients can undergo LPD procedure at high volume centers with rigorous selection and management. LPD could be a safe approach for liver cirrhotic patients with pancreatic and periampullary diseases compared with PD approach.

## Supporting information

**S1 Fig. Overall survival of cirrhotic patients and control patients according to different tumor.** A: Overall survival of cirrhotic patients and control patients with pancreatic carcinoma. B: Overall survival of cirrhotic patients and control patients with periampullary carcinoma.
(TIF)

**S1 Table. Tumor size according to different tumor between two groups.**
(DOCX)

## Acknowledgments

The authors declare no conflicts of interest.

## Author Contributions

**Conceptualization:** Ke Cheng, Bing Peng.

**Data curation:** Wei Liu, Jiaying You.

**Formal analysis:** Jiaying You, Xin Wang.

**Methodology:** Shashi Shah.

**Software:** Shashi Shah.

**Supervision:** Yunqiang Cai, Bing Peng.

**Validation:** Yunqiang Cai, Xin Wang.

**Writing – original draft:** Ke Cheng.

**Writing – review & editing:** Yunqiang Cai.

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
