## [Decision Letter · Decision Letter 0]

18 Nov 2020

PONE-D-20-23981

Safety of laparoscopic pancreaticoduodenectomy in patients with liver cirrhosis: a propensity score matching study

PLOS ONE

Dear Dr. Peng,

Thank you for submitting your manuscript to PLOS ONE. After careful consideration, we feel that it has merit but does not fully meet PLOS ONE’s publication criteria as it currently stands. Therefore, we invite you to submit a revised version of the manuscript that addresses the points raised during the review process.

Please revise accordingly.

We look forward to receiving your revised manuscript.

Kind regards,

Academic Editor

PLOS ONE

Journal Requirements:

2. Thank you for stating in the text of your manuscript ". All data, which were collected in May 2020, were fully anonymized before we accessed them." Please also add this information to your ethics statement in the online submission form.

3.Thank you for stating the following in the Acknowledgments Section of your manuscript:

[This work was supported by grants from the 1.3.5 project for disciplines of excellence-clinical Research Incubation Project, West China Hospital, Sichuan University (No.2018HXFH015).]

 [The author (s) received no specific funding for this work.]

Reviewers' comments:

Reviewer's Responses to Questions

**Comments to the Author**

1. Is the manuscript technically sound, and do the data support the conclusions?

Reviewer #1: Yes

Reviewer #2: Yes

2. Has the statistical analysis been performed appropriately and rigorously? 

Reviewer #1: I Don't Know

Reviewer #2: Yes

3. Have the authors made all data underlying the findings in their manuscript fully available?

Reviewer #1: Yes

Reviewer #2: Yes

4. Is the manuscript presented in an intelligible fashion and written in standard English?

Reviewer #1: No

Reviewer #2: No

5. Review Comments to the Author

Reviewer #1: There are a multitude of translation mistakes. I have highlighted some of them. The text should be read by a native English speaker.

On line 88 the authors say they excluded patients with severe malignant tumors? Please define severe malignant tumors.

On line 148 the authors define the types of pancreatic tumors in cirrhotic group. Please define and compare these tumors with ones in the control group.

Also the authors do not define the types of pancreas tumors were included in the study or what stage (TNM). Please specify as some can influence the overall survival no matter what surgical technique is used. This aspect needs to be taken into account when calculating the propensity score.

Also on line 88 the authors mention that cirrhosis was defined on 3 different techniques (CT, US, biopsy). Were all of these used on every patient? Were there patients diagnosed only with ultrasound?

The authors do not mention the cause of cirrhosis. Please specify as different forms can have different impact on survival.

Reviewer #2: The study is well done, it is a study with an importance in the pathology it serves (pancreatic pathology). The highlighting of the laparoscopic technique of duodenopancreatectomy in cirrhotic patients is a topic of current interest and deserves to be studied in detail. The English language should be revised throughout the manuscript because there are mistakes and phrases that do not make sense or do not express the content well. It should be stipulated in the manuscript whether or not the patients have signed the informed consent. From a bibliographic point of view, the bibliographic notes should be newer than 2010 and I would suggest adding more bibliographic notes. I suggest making the stipulated changes in order to publish the manuscript later.

6. PLOS authors have the option to publish the peer review history of their article (what does this mean?). If published, this will include your full peer review and any attached files.

Reviewer #1: No

Reviewer #2: No

---

## [Author Response · Author response to Decision Letter 0]

31 Dec 2020

Dear Prof.Chen and dear reviewers,

Re:Manuscript ID: PONE-D-20-23981 and Title: Safety of laparoscopic pancreaticoduodenectomy in patients with liver cirrhosis: a propensity score matching study

Thank you for your letter and for the reviewers’ comments concerning our manuscript entitled “Safety of laparoscopic pancreaticoduodenectomy in patients with liver cirrhosis: a propensity score matching study”. Those comments are all valuable and very helpful for revising and improving our paper, as well as the important guiding significance to our researches. We have studied comments carefully and have made correction which we hope meet with approval. Revised portion are marked in the paper. The main corrections in the paper and responds to the reviewer’s comments are as flowing:

Responds to the editor’s comments:

1.Response to comment: (Please ensure that your manuscript meets PLOS ONE's style requirements, including those for file naming.)

Response: We have revised our manuscript according to the PLOS ONE's style requirements.

2.Response to comment: (All data, which were collected in May 2020, were fully anonymized before we accessed them." Please also add this information to your ethics statement in the online submission form.)

Response: We have added it in the online submission form of ethics statement.

3.Response to comment: (We note that you have provided funding information that is not currently declared in your Funding Statement. However, funding information should not appear in the Acknowledgments section or other areas of your manuscript. We will only publish funding information present in the Funding Statement section of the online submission form.

[The author (s) received no specific funding for this work.]

Please include your amended statements within your cover letter; we will change the online submission form on your behalf.)

Response: We deleted the funding information in the acknowledgments section. We also included amended statements in our cover letter.

4.Response to comment: (Your ethics statement should only appear in the Methods section of your manuscript. If your ethics statement is written in any section besides the Methods, please delete it from any other section.)

Response: We deleted ethics statement besides the methods.

Responds to the reviewer’s comments:

Reviewer #1:

1.Response to comment:(There are a multitude of translation mistakes. I have highlighted some of them. The text should be read by a native English speaker.)

Response: We have modified the language expression in the manuscript by a language polishing service.

2.Response to comment:(On line 88 the authors say they excluded patients with severe malignant tumors? Please define severe malignant tumors.)

Response: We defined the severe malignant tumors as locally advanced, metastatic, or inoperable advanced cancer (can not be performed LPD).

3.Response to comment:(On line 148 the authors define the types of pancreatic tumors in cirrhotic group. Please define and compare these tumors with ones in the control group)

Response: We also compare these tumors in the control group (In the control group, 30 (54%) patients were diagnosed with pancreatic carcinoma, 20 (36%) hand periampullary carcinoma, 2 (4%) exhibited chronic pancreatitis whereas 4 (7%) patient had other benign or borderline pancreas tumors.).

4.Response to comment:(Also the authors do not define the types of pancreas tumors were included in the study or what stage (TNM). Please specify as some can influence the overall survival no matter what surgical technique is used. This aspect needs to be taken into account when calculating the propensity score.)

Response: In our study, all pancreatic carcinoma cases are Pancreatic ductal adenocarcinoma (PDAC). Periampullary carcinoma cases are defined as primary malignant tumors located in ampulla, duodenum, or common bile. We did not demonstrate the TNM stage and take TNM stage into account when calculating the propensity score for two reasons. Firstly, considering a relative long study period (October 2010 and December 2019), some early cases fail to record the lymphatic metastasis. Besides, the AJCC guideline revised many times in decade with changing definition in TNM stage, so we could not correctly assess the TNM stage. Secondly, the TNM stage assessment in pancreatic carcinoma and periampullary carcinoma are far different and we included many other diseases which were inapplicability in TNM stage. To sum up, we choose a more objective indicator, tumor sizes, to evaluate the tumor. Considering the complexity of different diseases, including some non-tumor cases, in our study and some cases are not applicable for TNM stage, we did not take TNM stage into calculating the propensity score. We compared the tumor sizes between cirrhotic group and control group according to different tumor. The result did not show significant difference between two groups (supplement 1).

Supplement 1. Tumor size according to different tumor between two groups.

 Patients with liver cirrhosis Control patients P-value

Tumor size (cm) 

Pancreatic carcinoma (n=45) 3.18±1.06 3.15±0.99 0.926

Periampullary carcinoma （n=29） 1.95±0.85 2.32±1.05 0.372

Indeed, the types of tumor can influence the overall survival. So we compare the tumor sizes between cirrhotic group and control group according to different kinds of tumor (supplement 2). The result also did not show significant difference between two groups.

Supplement 2. Overall survival of cirrhotic patients and control patients according to different tumor.

A: Overall survival of cirrhotic patients and control patients with pancreatic carcinoma. B: Overall survival of cirrhotic patients and control patients with periampullary carcinoma

 We think the advice of reviewer is very valuable. We will standardize our medical records data collection in the future study to compensate the fault.

5.Response to comment:(Also on line 88 the authors mention that cirrhosis was defined on 3 different techniques (CT, US, biopsy). Were all of these used on every patient? Were there patients diagnosed only with ultrasound?)

Response: Not all patients were diagnosed by all methods. The cirrhotic liver was diagnosed by any one of techniques (ultrasound, CT examination, intra-operative findings, liver biopsies ). 

6.Response to comment:(The authors do not mention the cause of cirrhosis. Please specify as different forms can have different impact on survival.)

Response: We have demonstrated the cause of cirrhosis in Table 2. 

Cause of liver cirrhosis 

HBV 13[46]

Alcohol abuse 4[14]

Biliary-oriental cirrhosis 7[25]

Schistosomiasis 1[4]

Cryptogenic cirrhosis 3[11]

We also noticed that different forms may have different impacts on survival, but sample sizes in some types are insufficient so we could not discuss the question in the current study. We think this question is very valuable and we hope we could discuss this question in further larger sample sizes study.

Reviewer #2:

1.Response to comment:(The English language should be revised throughout the manuscript because there are mistakes and phrases that do not make sense or do not express the content well.)

Response: We have modified the language expression in the manuscript by a language polishing service.

2.Response to comment:(It should be stipulated in the manuscript whether or not the patients have signed the informed consent)

Response: We demonstrated it in the manuscript (Patients signed the informed consent that their clinic data will be used in medical study before admitted to hospital.). 

3.Response to comment:(From a bibliographic point of view, the bibliographic notes should be newer than 2010 and I would suggest adding more bibliographic notes. I suggest making the stipulated changes in order to publish the manuscript later.)

Response: We have added the reference in the manuscript according to the reviewer’s note(35). Some old references aim to present previous opinions of liver cirrhosis in surgery. And we have renewed some old references(23,25,27,28).

Special thanks to you for your good comments.

Sincerely,

Dr.Peng

---

## [Decision Letter · Decision Letter 1]

7 Jan 2021

PONE-D-20-23981R1

Safety of laparoscopic pancreaticoduodenectomy in patients with liver cirrhosis using propensity score matching

PLOS ONE

Dear Dr. Peng,

Thank you for submitting your manuscript to PLOS ONE. After careful consideration, we feel that it has merit but does not fully meet PLOS ONE’s publication criteria as it currently stands. Therefore, we invite you to submit a revised version of the manuscript that addresses the points raised during the review process.

Please revise accordingly.

We look forward to receiving your revised manuscript.

Kind regards,

Academic Editor

PLOS ONE

Reviewers' comments:

Reviewer's Responses to Questions

**Comments to the Author**

1. If the authors have adequately addressed your comments raised in a previous round of review and you feel that this manuscript is now acceptable for publication, you may indicate that here to bypass the “Comments to the Author” section, enter your conflict of interest statement in the “Confidential to Editor” section, and submit your "Accept" recommendation.

Reviewer #1: All comments have been addressed

Reviewer #3: All comments have been addressed

2. Is the manuscript technically sound, and do the data support the conclusions?

Reviewer #1: Yes

Reviewer #3: Yes

3. Has the statistical analysis been performed appropriately and rigorously? 

Reviewer #1: Yes

Reviewer #3: Yes

4. Have the authors made all data underlying the findings in their manuscript fully available?

Reviewer #1: Yes

Reviewer #3: Yes

5. Is the manuscript presented in an intelligible fashion and written in standard English?

Reviewer #1: Yes

Reviewer #3: Yes

6. Review Comments to the Author

Reviewer #1: The authors have responded to the raised issues with the article. The paper is fit for publication from my point of view.

Reviewer #3: I have been given the task to review the revised manuscript in the light of comments made by my earlier worthy reviewers.

I commend you for a very important study you have carried out in a difficult terrain. Most of the concerns have been addressed. I have only a few comments:

1. In the survival data you have mentioned 25 (34%) as lost to follow up. It needs elaboration. how many of them were Cirrhotic and the remaining as non-cirrhotic matched. This is important because the cirrhotic patients are already 28, a smaller no.

2. This LOSS TO FOLLOW UP should be mentioned as a limitation in the discussion too.

3. The language and syntax although markedly improved, but still would require a final closer look. I noted FORM rather than the correct word FROM, 3rd line under DISCUSSION. similarly please correct CAN BE UNDERGO to CAN UNDERGO in CONCLUSIONS

4. The last statement under CONCLUSIONS reads: LPD could be a safe approach for managing liver cirrhosis in patients

296 compared with PD approach., Please clarify: Do you mean you can MANAGE cirrhosis with laparoscopic pancreatico duodenectomy or you can do LPD for malignancies IN CIRRHOTIC PATIENTS?

7. PLOS authors have the option to publish the peer review history of their article (what does this mean?). If published, this will include your full peer review and any attached files.

Reviewer #1: No

Reviewer #3: No

---

## [Author Response · Author response to Decision Letter 1]

8 Jan 2021

Dear Prof.Chen and dear reviewers,

Re:Manuscript ID: PONE-D-20-23981R1 and Title: Safety of laparoscopic pancreaticoduodenectomy in patients with liver cirrhosis using propensity score matching.

Thank you for your letter and for the reviewers’ comments concerning our manuscript entitled “Safety of laparoscopic pancreaticoduodenectomy in patients with liver cirrhosis using propensity score matching”. Your comments and those of the reviewers were highly insightful and enabled us to greatly improve the quality of our manuscript. In the following pages are our point-by-point responses to each of the comments of the reviewers.

Responds to the reviewer’s comments:

Reviewer #1: no response

Reviewer #3:

1. Response to comment:(In the survival data you have mentioned 25 (34%) as lost to follow up. It needs elaboration. how many of them were Cirrhotic and the remaining as non-cirrhotic matched. This is important because the cirrhotic patients are already 28, a smaller no.)

Response: We apologize for the ambiguous expressions in the manuscript. “lost during follow up” means patients died during follow up period rather than loss to follow up. We have revised the expression in the manuscript (A total of 25 (34%) patients died during follow-up. Of the 25 patients, 7 patients were in the cirrhotic group and 18 patients in the control group. ). 

2. Response to comment:(This LOSS TO FOLLOW UP should be mentioned as a limitation in the discussion too.)

Response: Actually, no patient loss to follow up in our study. It is an ambiguous expression in our manuscript and we have revised it. 

3. Response to comment:(The language and syntax although markedly improved, but still would require a final closer look. I noted FORM rather than the correct word FROM, 3rd line under DISCUSSION. similarly please correct CAN BE UNDERGO to CAN UNDERGO in CONCLUSIONS)

Response: We have rechecked and amended our manuscript. 

4. Response to comment: (The last statement under CONCLUSIONS reads: LPD could be a safe approach for managing liver cirrhosis in patients 296 compared with PD approach., Please clarify: Do you mean you can MANAGE cirrhosis with laparoscopic pancreatico duodenectomy or you can do LPD for malignancies IN CIRRHOTIC PATIENTS?)

Response: We have revised the expression in the manuscript (LPD could be a safe approach for liver cirrhotic patients with pancreatic and periampullary diseases compared with PD approach.). We mean LPD may be a more safe approach for cirrhotic patients.

Special thanks to you for your good comments. We shall look forward to hearing from you at your earliest convenience.

Sincerely,

Dr.Peng

---

## [Decision Letter · Decision Letter 2]

11 Jan 2021

PONE-D-20-23981R2

Safety of laparoscopic pancreaticoduodenectomy in patients with liver cirrhosis using propensity score matching

PLOS ONE

Dear Dr. Peng,

Thank you for submitting your manuscript to PLOS ONE. After careful consideration, we feel that it has merit but does not fully meet PLOS ONE’s publication criteria as it currently stands. Therefore, we invite you to submit a revised version of the manuscript that addresses the points raised during the review process.

Please address the issues uncovered by the reviewers and revise accordingly.

We look forward to receiving your revised manuscript.

Kind regards,

Academic Editor

PLOS ONE

Reviewers' comments:

Reviewer's Responses to Questions

**Comments to the Author**

1. If the authors have adequately addressed your comments raised in a previous round of review and you feel that this manuscript is now acceptable for publication, you may indicate that here to bypass the “Comments to the Author” section, enter your conflict of interest statement in the “Confidential to Editor” section, and submit your "Accept" recommendation.

Reviewer #1: All comments have been addressed

Reviewer #3: (No Response)

2. Is the manuscript technically sound, and do the data support the conclusions?

Reviewer #1: Yes

Reviewer #3: (No Response)

3. Has the statistical analysis been performed appropriately and rigorously? 

Reviewer #1: Yes

Reviewer #3: (No Response)

4. Have the authors made all data underlying the findings in their manuscript fully available?

Reviewer #1: Yes

Reviewer #3: (No Response)

5. Is the manuscript presented in an intelligible fashion and written in standard English?

Reviewer #1: Yes

Reviewer #3: (No Response)

6. Review Comments to the Author

Reviewer #1: The authors have adresed the raised issues and the article is fit for publication from my personal point of view.

Reviewer #3: Thank you very much for promptly responding/ rectifying to the queries raised in the manuscript.

I have noticed with interest the correction made for LOSS to FOLLOW up revised as DIED! But then the next sentence needs elaboration. What do you mean by writing"significantly different" from control group. Please explain the Fig. 1 and survival data of both groups clearly. In the Fig. the last half of the curve is the same for both the groups. In the table you have mentioned 90 mortality as 25% in cirrhotic patients and 5% in control group. That means significantly more in cirrhotic patients (expected and understandable).

7. PLOS authors have the option to publish the peer review history of their article (what does this mean?). If published, this will include your full peer review and any attached files.

Reviewer #1: No

Reviewer #3: No

---

## [Author Response · Author response to Decision Letter 2]

13 Jan 2021

Dear Prof.Chen and dear reviewers,

Re:Manuscript ID: PONE-D-20-23981R2 and Title: Safety of laparoscopic pancreaticoduodenectomy in patients with liver cirrhosis using propensity score matching.

Thank you for your letter and for the reviewers’ comments concerning our manuscript entitled “Safety of laparoscopic pancreaticoduodenectomy in patients with liver cirrhosis using propensity score matching”. Your comments and those of the reviewers were highly insightful and enabled us to greatly improve the quality of our manuscript. In the following pages are our point-by-point responses to each of the comments of the reviewers.

Responds to the reviewer’s comments:

Reviewer #1: no response

Reviewer #3:

1. Response to comment: (What do you mean by writing"significantly different" from control group. Please explain the Fig. 1 and survival data of both groups clearly. In the Fig. the last half of the curve is the same for both the groups. )

Response: During the language polishing, the sentence was changed. We writed “no significantly different” but the “no” is missed in the last version. Sorry for the fault. Actually, the two groups have no significant difference in OS though Kaplan-Meier analysis (P=0.991). 

2. Response to comment: ( In the table you have mentioned 90 mortality as 25% in cirrhotic patients and 5% in control group. That means significantly more in cirrhotic patients (expected and understandable).)

Response: We compared the 90-day mortality between cirrhotic group and control group. Although cirrhotic group have higher mortality (14%) compared with control group (5%), but the result is no significantly different (P=0.163). 

 Patients with liver cirrhosis [%] (n=28) Control patients [%] (n=56) P-value

Hospital mortality 1[4] 1[2] 0.613

90-day mortality 4[14] 3[5] 0.163

This is a part of our table, more details can be found in Table 4.

Special thanks to you for your good comments. We shall look forward to hearing from you at your earliest convenience.

Sincerely,

Dr.Peng

---

## [Decision Letter · Decision Letter 3]

19 Jan 2021

Safety of laparoscopic pancreaticoduodenectomy in patients with liver cirrhosis using propensity score matching

PONE-D-20-23981R3

Dear Dr. Peng,

We’re pleased to inform you that your manuscript has been judged scientifically suitable for publication and will be formally accepted for publication once it meets all outstanding technical requirements.

Kind regards,

Academic Editor

PLOS ONE

Additional Editor Comments (optional):

Reviewers' comments:

Reviewer's Responses to Questions

**Comments to the Author**

1. If the authors have adequately addressed your comments raised in a previous round of review and you feel that this manuscript is now acceptable for publication, you may indicate that here to bypass the “Comments to the Author” section, enter your conflict of interest statement in the “Confidential to Editor” section, and submit your "Accept" recommendation.

Reviewer #1: All comments have been addressed

Reviewer #3: (No Response)

2. Is the manuscript technically sound, and do the data support the conclusions?

Reviewer #1: Yes

Reviewer #3: (No Response)

3. Has the statistical analysis been performed appropriately and rigorously? 

Reviewer #1: Yes

Reviewer #3: (No Response)

4. Have the authors made all data underlying the findings in their manuscript fully available?

Reviewer #1: Yes

Reviewer #3: (No Response)

5. Is the manuscript presented in an intelligible fashion and written in standard English?

Reviewer #1: Yes

Reviewer #3: (No Response)

6. Review Comments to the Author

Reviewer #1: The authors have amended the issues with the article through the multiple previous reviews, as such the article is qualified for publication.

Reviewer #3: Thanks for addressing the issues raised in the last submission. The necessary amendments have been done in the resubmitted version of the manuscript. I have gone through the revisions and find the manuscript good now.

7. PLOS authors have the option to publish the peer review history of their article (what does this mean?). If published, this will include your full peer review and any attached files.

Reviewer #1: No

Reviewer #3: No

---

## [Editor Report · Acceptance letter]

21 Jan 2021

PONE-D-20-23981R3 

Safety of laparoscopic pancreaticoduodenectomy in patients with liver cirrhosis using propensity score matching 

Dear Dr. Peng:

I'm pleased to inform you that your manuscript has been deemed suitable for publication in PLOS ONE. Congratulations! Your manuscript is now with our production department. 

Kind regards, 

on behalf of

Dr. Robert Jeenchen Chen 

Academic Editor

PLOS ONE